# Effect of Linear and Nonlinear Pedagogy Physical Education Interventions on Children’s Physical Activity: A Cluster Randomized Controlled Trial (SAMPLE-PE)

**DOI:** 10.3390/children8010049

**Published:** 2021-01-15

**Authors:** Matteo Crotti, James R. Rudd, Simon Roberts, Lynne M. Boddy, Katie Fitton Davies, Laura O’Callaghan, Till Utesch, Lawrence Foweather

**Affiliations:** 1Research Institute for Sport and Exercise Sciences, Liverpool John Moores University, Liverpool L2 2QP, UK; M.Crotti@2016.ljmu.ac.uk (M.C.); J.R.Rudd@ljmu.ac.uk (J.R.R.); S.Roberts2@ljmu.ac.uk (S.R.); L.M.Boddy@ljmu.ac.uk (L.M.B.); or ad5858@coventry.ac.uk (K.F.D.); L.A.OCallaghan@2017.ljmu.ac.uk (L.O.); 2Centre of Sport, Exercise and Life Sciences, Coventry University, Coventry CV1 5FB, UK; 3Department of Pedagogical Assessment and Potential Development, Institute of Educational Sciences, University of Münster, 48149 Münster, Germany; till.utesch@uni-muenster.de

**Keywords:** teaching, curriculum, primary school, accelerometers, movement competence

## Abstract

Background: School-based interventions are a key opportunity to improve children’s physical activity (PA); however, there is lack of evidence about how pedagogical approaches to motor learning in physical education (PE) might affect PA in children. Therefore, this study aimed to assess how different pedagogical approaches in PE might affect children’s PA. Methods: Participants (*n* = 360, 5–6 years) from 12 primary schools within the SAMPLE-PE randomized controlled trial were randomly allocated to either Linear Pedagogy (LP: *n* = 3) or Nonlinear Pedagogy (NP: *n* = 3) interventions, where schools received a 15-week PE intervention delivered by trained coaches, or to a control group (*n* = 6), where schools followed usual practice. ActiGraph GT9X accelerometers were used to assess PA metrics (moderate-to-vigorous PA, mean raw acceleration and lowest acceleration over the most active hour and half hour) over whole and segmented weeks at baseline, immediately post-intervention and 6 months follow-up. Intention to treat analysis employing multilevel modelling was used to assess intervention effects. Results: LP and NP interventions did not significantly affect children’s PA levels compared to the control group. Conclusion: PE interventions based on LP and NP alone might not be effective in improving habitual PA in children.

## 1. Introduction

Increased physical activity (PA) in children is associated with positive effects on quality of life [1], self-perception [2], cardiovascular fitness [3], metabolic function [4] and cognition [5]. Children who are physically active are also more likely to become healthy and active adults [6]. Despite these benefits, a large number of children across the globe do not engage in the recommended guidelines of 60 min of moderate-to-vigorous physical activity (MVPA) per day for healthy growth and development [7,8,9,10,11], with children from areas of deprivation participating in even lower levels of PA [12]. In view of this, a global call of action was raised to increase PA in children using interventions that could be feasibly implemented at scale [13].

School is considered an ideal setting to promote current and future PA on a population level as large numbers of children can be reached [14,15]. A recent review by Grao-Cruces et al. (2020) reported that children spend, on average, between 14 and 61 min in MVPA at school, showing that children engage in a considerable amount of PA and can even meet PA guidelines during the school day [16]. Physical education (PE) is a key occasion for children to engage in MVPA during school time, with evidence suggesting that children are more physically active during school days, including PE, than during other school days [17]. PE is not merely an opportunity for children to engage in PA, it is widely recognized as playing a crucial role in the development of knowledge and skills to foster their PA engagement throughout life [15,17,18,19]. Despite this, there is weak evidence and limited understanding about how learning experiences in PE affect children’s PA during school time and outside of school [20,21]. Studies reporting a positive effect of PE interventions on children’s habitual PA have mostly measured PA using self-report or parent proxy questionnaires [22,23,24,25,26,27,28,29,30,31]. Self-reported or parent reported PA measurements in children are exposed to risk of bias such as recall and social desirability bias together with the difficulties children have in recognizing different PA levels and constructs [32]. Therefore, future studies should assess the effect of PE interventions on PA using device-based measurements, such as accelerometers [21].

International and national PE curriculum guidelines state that PE should support young children’s development of movement competence [19,33,34,35]. Movement competence is hereby defined as an individual’s degree of proficiently performing a broad range of movement skills, which also affects the underlying mechanisms, including quality of movement, motor coordination and motor control [36]. Evidence indicates a positive and reciprocal association between movement competence and PA engagement in children, with children possessing high movement competence being more likely to engage in PA during their adolescence and adulthood [37,38,39,40,41]. Thus, learning foundational skills, such as catching, bouncing a ball, swimming, leaping, cycling and kicking, could enhance children’s actual and perceived capability to engage in PA, sport and recreational opportunities, positively affecting their PA levels [42]. While several studies have examined associations between movement competence and PA, there is limited evidence about how the quality of movement learning experienced through PE influences participation in PA [21,41,43]. PE pedagogical approaches underpinned by motor learning theories from contrasting standpoints, such as linear pedagogy and nonlinear pedagogy might affect both movement competence development as well as PA engagement [21,43,44]

Linear pedagogy is based on the Information Processing Theory [45] about learning. Information processing theory explains how specific inputs (sensory inputs and desired movement outcomes) experienced by learners are elaborated together with previous experiences before commencing the action and during the action based on sensory feedback to produce specific movement outcomes, leading to learning outcomes (schemas and skill learning) [46]. From this perspective, providing a set of movement experiences of increasing difficulty should lead to a linear learning progression through cognitive stages (cognitive, associative, autonomous), with improving movement proficiency accompanied by a reduction in cognitive processing while performing [46]. Linear pedagogy can be characterized by a teacher-centered approach to PE, as (a) children should learn the optimal movement patterns for each movement skill and all children should conform to these idealistic movement patterns; (b) movement skills should be broken down into basic and simpler movements to facilitate learning; (c) movement variability within a task is seen as detrimental for learning and therefore should be reduced; (d) teachers in early learning should encourage an internal focus of attention in children who are performing skills to reduce cognitive load, while, as children become proficient in the skill, teachers would encourage an external attention of focus [46,47]. While the characteristics of linear pedagogy are comparable with traditional practices in PE that follow a sport-as-technique approach [48], linear pedagogy is based on motor learning theory and should, therefore, lead to more beneficial outcomes than atheoretical approaches currently employed [49,50]. With teacher-led, linear approaches, the development of motor proficiency in one optimal technique may result in fast learning, leading to early feelings of success that should increase perceptions of competence, contributing to higher levels of motivation in the lesson, as well as PE and PA more broadly [51,52].

Nonlinear pedagogy has been developed and constructed based upon an ecological dynamics approach. At the heart of this pedagogical framework is exploratory learning, with an emphasis on encouraging individualized movement solutions [53]. From this perspective, providing children with the freedom to explore a carefully designed learning environment will lead to constraint led synergy formation that will result in the performance of functional movement solutions [54]. Consequently, Nonlinear pedagogy involves a child-centered PE approach where teachers channel children’s learning by modifying task constraints to assist in the synergy formation of skills that will be functional to the task at hand. A key aspect of this is not to over constrain synergy formation, as such, the manipulation of equipment or rules of a game would be preferred over providing the child with direct instruction [53]. For teachers delivering a nonlinear pedagogical approach, movement skills should be practiced in representative environments where perception and action are not broken. This means learning activities should be situated in performance contexts that capture the dynamics where the skills to be learnt can be performed, developed and acquired. In a nonlinear pedagogy approach, teachers modify individual, task and environmental constraints to support exploration and with reference to nonlinearity in learning, variability is seen as inherently present in how movement is controlled and produced. Variability in movement control can thus be functional and is to be encouraged. Lastly, in nonlinear pedagogy, teachers should encourage an external focus of attention to support self-organization. Several authors have proposed that nonlinear pedagogy could support children’s basic psychological needs of autonomy, relatedness and competence from a self-determination theory perspective and, therefore, could lead to higher levels of motivation towards PA engagement that might positively affect PA levels in children compared to traditional teaching approaches [44,55,56].

In summary, primary (elementary) PE is an important setting for PA promotion and child development, especially for children from areas of high deprivation who participate in less PA compared to children from more affluent areas. Movement competence is an important outcome of PE and enhanced learning experiences in PE, based on motor learning theory, could lead to greater engagement in PA compared to atheoretical approaches used in current practice. To the best of our knowledge, no study has examined the effect of linear and nonlinear pedagogy PE interventions on children’s habitual PA and, more broadly, there is a lack of evidence concerning how interventions aimed at improving movement competence might affect children’s PA [21,41,43]. Therefore, the aim of this study was to evaluate the effect of linear and nonlinear pedagogy PE interventions on the PA levels of 5–6-year-old children from areas of high deprivation.

## 2. Materials and Methods

### 2.1. Study Design and Participants

Ethical approval for the study was granted by the University Research Ethics Committee (Reference 17/SPS/031). This study formed part of the wider SAMPLE-PE project—a registered (ClinicalTrials.gov identifier: NCT03551366) cluster randomized controlled trial evaluating the effect of PE pedagogical approaches guided by motor learning theories on 5–6-year-old children’s physical literacy [54]. The main trial methods of the study have been described in detail elsewhere [54]. Briefly, primary schools from deprived areas in the North West of England were contacted and invited to take part in the study (Figure 1). The head-teachers of 12 primary schools provided informed consent to participate in the SAMPLE-PE project. Subsequently, Year 1 children (5–6 years old) within the participating schools were invited to take part in the study and parental informed consent, together with child assent to participate in the study, were collected. The children who did not provide consent to participate in the research study took part in physical education lessons both in the intervention and control groups. Children who could not take part in PE because of medical conditions, severe learning disabilities or special educational needs were not eligible to take part in the research.

The 12 schools were randomly allocated to a nonlinear pedagogy intervention group (3 schools), a linear pedagogy intervention group (3 schools), or a control group (6 schools). Baseline data (T0) collection occurred in January–February 2018. The intervention started immediately after baseline assessments and consisted of two PE lessons per week for 15 weeks, delivered by trained coaches. Control group schools were asked to provide their usual PE practice for two lessons per week during the same period. Post-intervention assessments (T1) were completed within 2 weeks after the intervention period between June and July 2018, while follow-up assessments (T2) took place 6 months after post-intervention assessments between January and early March 2019. The design, conduct and reporting of this study was designed in accordance with the Consolidated Standards of Reporting Trials (CONSORT) [57].

### 2.2. Intervention

#### 2.2.1. Intervention Deliverers Training

Given that most of the generalist primary school teachers lack the confidence and competence to effectively teach PE [58], coaches were recruited to deliver the linear and nonlinear pedagogy PE interventions. This in line with current practice in primary PE in England where the majority of primary schools currently employ sports coaches from external providers to deliver PE [59]. Intervention deliverers (coaches) were recruited from a university in-house coaching provider and within the research team. Coaches were required to hold a Level 2 UK coaching qualification in any sport. All of the PE coaches recruited in the project were enrolled into a training program to deliver either the linear or the nonlinear pedagogy intervention. Before assigning the coaches to one of the training programs, members of the research team observed the coaches while delivering a PE lesson in a primary school not participating in the project. Subsequently, based on the observed lessons, the researchers assigned the coaches to the pedagogical approach training (linear or nonlinear) more aligned to their teaching practices. The decision to assign coaches based on their alignment with intervention pedagogies was made to maximize the likelihood of intervention fidelity.

Three coaches were assigned to the nonlinear pedagogy curriculum training, while two coaches received the linear pedagogy curriculum training. The training consisted of one session a week for 5 weeks, delivered by a member of the research team. Each session lasted 3 h, divided evenly into theory and practice. Practical sessions were carried out with Year 2 children (6–7 years old) from a primary school that was not involved in the randomized controlled trial. At the end of the training, each PE coach received a scheme of work and lesson plans designed by the research team in collaboration with them outlining the content of PE lessons to guarantee consistency in the intervention content delivery. Furthermore, coaches received a pedagogical framework and a resource pack about delivering either a linear or nonlinear pedagogical approach. Additionally, the material used during training sessions, together with the recordings of the sessions, were made available for the coaches online. Following the training, coaches were supported by the research team through weekly telephone calls to discuss the design and delivery of lessons and to assist in adapting lesson plans to their intervention classes.

#### 2.2.2. Interventions

The SAMPLE-PE interventions are described in detail within the study protocol [54]. Briefly, the main aim of the wider SAMPLE-PE project was to assess the effect of linear and nonlinear pedagogies in fostering physical literacy among 5–6-year-old children from deprived areas of North-West England. Given that linear and nonlinear pedagogies are based on motor learning theories, the primary outcome in the SAMPLE-PE project was movement competence. Both linear and nonlinear pedagogy interventions lasted 15 weeks and comprised thirty PE lessons, which were divided into three content blocks of five weeks, corresponding to 10 lessons each focusing on “Dance”, “Gymnastic” and “Object control skills” as overarching themes. The overarching themes of each PE lesson specified in the intervention deliverers’ scheme of work (e.g., “Fast and slow movements” in a Dance lesson, “Rolling” in a Gymnastic lesson, “Underarm throw” in an object-control lesson) were the same for both linear and nonlinear pedagogy interventions to minimize content differences between linear and nonlinear curriculums. Intervention duration was chosen based on previous literature, showing that interventions lasting between 6 and 15 weeks are effective in increasing children movement competence [60,61].

#### 2.2.3. Linear Pedagogy Intervention

The well-established principles and theories of direct instruction were used by the research team and trained PE coaches to guide the design of the linear intervention [49]. Consequently, linear pedagogy PE lessons generally followed a traditional structure involving: (1) a warm-up activity, (2) practicing passive movement skills within drills, (3) a performance or game activity to apply the movement skills learnt during the lesson and (4) a cool down. Coaches were asked to provide clear instructions and demonstrations to the children before each task, and to provide augmented corrective feedback during each activity. Emphasis was given to executing and reiterating movement skills in a desired performance or outcome as previously demonstrated by the coach. Coaches used the principles from Gentile’s taxonomy and challenge point framework [62,63] to create progressions of tasks of increasing difficulty from simple and controlled movements to complex and dynamic actions. Coaches were trained to evaluate the children’s progression in movement skills proficiency using Fitts and Poster’s cognitive stages (cognitive, associative, autonomous) [46] and to adapt the difficulty of the tasks based on children’s skill level.

#### 2.2.4. Nonlinear Pedagogy Intervention

Nonlinear pedagogy theories and principles were used by the research team and trained PE coaches to guide the design of the nonlinear intervention [57]. Specifically, the research team together with the coaches delivering the intervention identified relevant constraints to design PE lessons, including environmental (e.g., space boundaries, equipment type, equipment number, spatial organization of objects), task (e.g., activity type, rules within a task, duration of the task, number of participants) and individual constraints (e.g., age, sex, socioeconomic demographic). At the beginning of each lesson, coaches invited children to explore the PE hall and the different objects within the environment (e.g., benches, mats, hoops, cones). The lesson continued with activities representative of game, sport or performance situations where coaches introduced variability by changing constraints. Coaches used the Space Task Equipment People (STEP) framework to identify and manage constraints within the lessons [64]. Furthermore, coaches were trained to monitor children’s progress in movement learning using Newell’s stages of motor learning (coordination, control and skill) and to modify and individualize constraints based on children’s motor learning stage [64]. Coaches did not provide demonstrations or feedback during activities. Alternatively, they invited children to reflect using questioning strategies or to observe their peers. Coaches also used questioning to foster an external focus of attention in the child to infuse variability in the task and channel children learning (e.g., How could we make this task more difficult? How can your teammates help you in this task? How many ways to move on the floor can you think about?).

### 2.3. Outcomes

Demographic outcomes were collected during baseline data collection (January–February 2018) while anthropometric and physical activity outcomes were collected during each data collection point, comprising baseline, post-intervention (June–July 2018) and follow-up (January–early March 2019).

#### 2.3.1. Demographics

Information about children’s demographics (i.e., date of birth, gender, ethnicity, home postcode and special educational needs) were provided by parents or guardians within a questionnaire that was returned with the consent form. Household postcode was used to classify children into deciles of deprivation level using the English indices of deprivation [65].

#### 2.3.2. Anthropometrics

Children’s anthropometric measurements took place within the schools. Stature (The Leicester Height Measure, Child Growth Foundation, Leicester, UK), to the nearest 0.1 cm, and mass (model 760, Seca, Hamburg, Germany), to the nearest 0.1 kg, were measured using standard procedures [66]. All measurements were taken twice, while a third measurement was taken if the first two differed by more than 1%. Body Mass Index was calculated and converted to standardized Body Mass Index (BMI) z-scores using the International Obesity Task Force (IOTF) classification [67].

#### 2.3.3. Physical Activity

PA was assessed using ActiGraph GT9X accelerometers (ActiGraph, Pensacola, FL, USA) set to record accelerations at 30 Hz over 1 s epochs within a range of ±8 g on the x, y and z axes. Children were asked to wear a GT9X accelerometer on their nondominant wrist for an entire week and to only remove the device when having a bath, swimming, or for safety reasons. Furthermore, children were encouraged to wear the monitor all day including sleeping hours, and to bring the device back to school on a specific date (i.e., 7 days after receiving it). Each participant received an accelerometer directly from a trained researcher within their school together with an information pack for the parents or guardians including a wear time diary and information about when to return the device to the school. Parents or guardians were asked to fill in the diary and record times when their child took off the device as well as the time when the child went to sleep and woke up. Where children did not wear the device for at least 3 weekdays and one weekend day for 10 valid hours, they were invited to wear the device again for an entire week. Teachers were asked to report to the research team whether the school had organized any special sport or activity events during school time during each measurement period that was not part of the normal school week and could disrupt children’s regular PA engagement patterns.

Following previous studies [68,69], children’s awake time was established as a standard period between 06:00 and 23:00, as the majority of the children did not wear the monitor during the night time. Consequently, sleep time was established as a standard period between 23:00 and 06:00 and all PA analysis included awake time only. The classification of valid wear time was done following the GGIR package [70] from R software Version 4.0.2 (www.r-project.org) default option over blocks of 15 min where each block was classified as nonwear time when the standard deviation of the 60 min interval around the block was less than 13 mg in at least 2 of the 3 axes or if the value range for at least 2 of the 3 axes was less than 50 mg [70]. A day of measurement was considered valid only when the participant had at least 10 h of valid wear time during waking hours while a measured week was considered valid when the participant was assessed over at least 3 valid week days and 1 valid weekend day [71]. Children’s PA levels during nonwear time were imputed based on recordings from other days as a default in the GGIR package [70]. In cases where children were re-monitored, the valid days from the first monitoring session and the re-monitoring session within the same assessment point (e.g., baseline) were pooled together. Only PA data from valid days within valid weeks were included in the final analysis. Furthermore, mean rainfall, mean temperature and day length, specific to the valid PA data, were obtained from the “Metoffice” [72] and “Timeanddate” websites [73] to account for seasonal variation in PA outcomes across each time point.

Raw acceleration data were downloaded from accelerometers in 1 s epochs and exported as .csv files using ActiLife software version 6.11.9 (ActiGraph, Pensacola, FL, USA). Raw data were then transformed into Euclidean Norm Minus One (ENMO) acceleration data using the GGIR package [70] from R software Version 4.0.2 (www.r-project.org). Subsequently, the GGIR package was used to compute mean ENMO acceleration, the minimum acceleration value within the most active hour of the day (M60), the minimum acceleration value within the most active half an hour of the day (M30) [74,75], together with time spent in MVPA based on age-appropriate cut-points [76]. We included mean ENMO, M60 and M30 as PA metrics in view of recent calls to use cut-point free metrics to facilitate the comparison of PA outputs from different brands of accelerometers and also to get a deeper insight on children’s PA engagement beyond MVPA [74]. Mean ENMO acceleration differs from MVPA as it represents the magnitude of total PA accumulated during the recording time and was found to be positively associated with health related outcomes in children [77]. M60 was chosen a PA metric for whole week and weekend as children are meant to engage in at least 60 min of MVPA per day and M60 provides valuable information about how active children were in their most active 60 min in a day. Following a similar rationale, we included M30 to assess PA within school time and outside of school in accordance with UK targets for primary school children to engage in 30 min of MVPA in school and 30 min of MVPA outside of school to achieve the recommended daily 60 min of MVPA [78,79]. Furthermore, M30 was found to be associated with health related outcomes comprising BMI, waist-to-height ratio and cardiorespiratory fitness in children [74].

### 2.4. Intervention Fidelity

The research team developed a checklist to assess the fidelity of the intervention through the video analysis of recorded PE lessons (Appendix A). The checklist included 9 items comprising 7 motor-learning-related items and 2 global items. Each item was scored on a 1 to 5 Likert sale, where a score of 1 corresponded to the observation being in line with linear pedagogy, while a score of 5 corresponded to the observation being in accordance with nonlinear pedagogy. Each motor learning related item was assessed 4 times within each lesson (once for each quartile of the PE lessons), while global items were assessed only once per lesson observed. Two independent researchers that were blinded to the group allocations were trained to code the lessons. The training consisted of: (1) reading specific literature concerning linear and nonlinear pedagogy, (2) reading the fidelity checklist, (3) consulting the research team about doubts concerning the checklist, (4) independently coding 2 PE lessons, (5) consulting a pedagogy expert to check the coded lessons and clarify any doubts, (6) collaborating to assess 6 PE lessons, (7) independently assessing 6 lessons and then comparing the results. The coders then assessed fidelity using the Fidelity Checklist within a total of 13 randomly selected PE lessons from the linear pedagogy, nonlinear pedagogy and control groups. Raters demonstrated high inter-rater reliability with an Intraclass Correlation Coefficient (ICC) equal to or higher than 0.97.

### 2.5. Randomization and Power

The participating schools were matched by number of students enrolled and then they were randomly allocated to either intervention or control groups using a computer-based algorithm. As a result, more schools were allocated to the control group to account for the higher risk of drop out, as a consequence of not receiving the intervention. The study was powered as reported in the SAMPLE-PE project protocol paper [54] to assess movement competence change in 3 groups over 3 time points with 90% power at a level of *p* < 0.05, adjusting for clustering at class level and allowing a dropout at each time point equal to 20%. As a result, the initial sample calculation aimed to recruit at least 314 participants.

### 2.6. Data Analysis

All data analysis was carried out using R Software (Version 4.0.2, www.r-project.org) and RStudio Software (Version 1.3.1056, www.rstudio.com). The main effect of time (the change from one timepoint to the next, averaged across groups), group (i.e., the difference between groups averaged across timepoints) and group by time interaction effects (the extent to which the difference between intervention and control groups is different at different timepoints) in children’s PA variables comprising MVPA, mean ENMO, M60 and M30 were assessed using multilevel linear regression models. Separate multilevel models were conducted to examine PA variables during whole week (habitual PA), weekend, school time (9 a.m. to 3 p.m.) and outside school (3 p.m. to 11 p.m.) during weekdays. Models considering the nested data structure were selected to maximize model fit assessed using the Chi-squared test while minimizing the complexity of the final model. Overall, observations (level 1) were nested within children (level 2) in multilevel models concerning whole week, weekend and week time outside school PA variables, as nesting by class (level 3) or school (level 4) did not increase the model fit or led to overfitting. Conversely, observations were nested within children and class in all multilevel models concerning school time PA variables as nesting by children and by class led to the best model fit. Based on previous studies identifying PA correlates, all models were adjusted for decimal age [80], sex [81,82], International Obesity Task Force (IOTF) BMI z-score [83], special educational needs [67], ethnicity [12], school sport events [84] and household neighborhood deprivation decile [65]. Furthermore, models were adjusted for accelerometer valid wear time [85], mean rainfall, mean temperature and day length [86,87], specific to the time of the week considered in the model. Based on published guidelines about dealing with missing data in randomized trials, we imputed missing data using the Multiple Imputation method [88,89]. We then performed both an intention to treat analysis on imputed data and a complete cases analysis as a sensitivity analysis [88,89]. Complete cases analysis was conducted in order to examine whether between-group effects differed from the intention to treat data analysis. Missing data (see Appendix A for details) were imputed applying Multiple Imputation method using the “mice” Package, employing the “Jomoimpute” function [90] within R software. A specific imputation was performed for each multilevel model comprising all the variables to be included in the model, accounting for multilevel nesting together with time by group interaction and creating 10 datasets of imputed data [91]. Separate multilevel models were run using each of the imputed datasets and then the estimates from the models were pooled [91,92]. The same multilevel linear regression model methods were also used to analyze the dataset without imputation.

## 3. Results

Figure 1 shows the flow of schools and participants through the trial. In total, 12 schools participated in the study (10% response rate). Schools that declined to participate provided diverse reasons for not taking part (e.g., too busy; already in receipt of external projects). Of the 410 potentially eligible children at T0 (baseline), 360 children were enrolled into the study (88% response rate) and 307 children (85% of participants) had valid PA data at either baseline, post-intervention and/or follow-up. Reasons for missing data included children being absent on data collection days, leaving school, declining to undertake measurement procedures, losing accelerometers, or not meeting the PA inclusion criteria (see Appendix A). Participant retention in the study from baseline to follow-up was 98%, 95% and 87% for the linear pedagogy, nonlinear pedagogy and control groups, respectively, with a larger proportion of control group children leaving school within the study period.

### 3.1. Baseline Characteristics

Table 1 shows the demographic and baseline characteristics of the study sample by group. The pooled sample comprised 360 children (55% girls) with a mean age of 5.9 (Standard Deviation (SD) = 0.3) years; 56% of the children were white British, while 44% were from other ethnicities; 12% reported special educational needs of mild and moderate severity and the vast majority lived in highly deprived areas, with 85% of the children living in areas amongst the 30 percent most deprived in England. Based on the International Obesity Task Force (IOTF) classifications, 17% of children were overweight and 6% were obese, while BMI was not assessed in 12% of children due to school absences. Of the 262 children with valid baseline PA data, 65%, 71% and 51% engaged in an average of 60 or more minutes of MVPA during the whole week, weekdays and weekends, respectively. Descriptive statistics concerning child characteristics in all outcome measures by group at baseline, post-intervention and follow-up assessments can be found in Appendix A.

### 3.2. Fidelity Assessment

Table 2 reports means and standard deviations of the pedagogy fidelity assessment. Nonlinear pedagogy intervention fidelity scores were all higher than 4, apart from category 4, which presented a score equal to 3.95. Linear pedagogy intervention fidelity scores were all lower than 1.77, while the control group scores ranged from 1.44 and 2.50. Given that scores of 1 and 2 on the Likert scale correspond to the observation being more in line with linear pedagogy and scores of 4 and 5 correspond to the observation being in line with nonlinear pedagogy, the fidelity check observation data indicated that linear and nonlinear interventions were delivered in line with their respective pedagogical principles. The control group presented characteristics that indicated closer alignment towards linear pedagogy principles.

### 3.3. Intervention Effect on Physical Activity Outcomes

The full outputs from the 24 multilevel models, including covariates, can be found in Appendix A (intention to treat analysis) and Appendix A (complete case analysis). Table 3 and Table 4 present model summaries in relation to intervention effects. The intention to treat analysis involved imputed data from all 360 children with a total of 1080 complete observations in each variable. There were no significant groups by time interaction effects in all the PA outcomes, inclusive of MVPA, mean ENMO, M60 and M30 for both whole week and weekend periods (see Table 3). As shown in Table 4, no significant group by time effects were observed for PA outcomes during school time (09:00 to 15:00) and outside of school (15:00 to 23:00) during weekdays. No group effects (i.e., the difference between groups using data averaged across T0, T1 and T2) were observed, apart from the linear pedagogy group, presenting lower M30 (β = −45.45 mg, SE = 22.58 mg, *p* = 0.045) compared to the control group within school time. For time effects (i.e., the change from one timepoint to the next, averaged across groups), it was observed that MVPA and mean ENMO decreased at follow-up during the weekend only.

The multilevel models’ complete case analysis involved data from 274 children with a total of 575 observations in each variable (53.2% of observations; see Appendix A for missing data information). Group by time interaction effects from the complete case analysis were largely consistent with the intention to treat analysis, with some exceptions (see Table 3 and Table 4). Specifically, at post-intervention (T1), a significant group by time interaction effect was found for the linear pedagogy interventions on MVPA and mean ENMO out of school weekday PA metrics, with negative intervention effects observed relative to the control group (MVPA: Estimate (β) = −7.74 min, Standard Error (SE) = 3.71 min, *p* = 0.037; mean ENMO: β = −12.24 mg, SE = 5.89 mg, *p* = 0.038). No significant group by time interaction effects were found for out of school weekday PA metrics at follow-up. At follow-up (T2), a significant group by time interaction effect was found for nonlinear pedagogy for MVPA in school, indicating a positive intervention effect compared to the control group (β = 5.18 min, SE = 2.11 min, *p*= 0.014). No group effects were observed, apart from the linear pedagogy intervention group, presenting, on average, higher MVPA (β = 4.85 min, SE = 2.26 min, *p* = 0.032), mean ENMO (β = 8.45 mg, SE = 3.59 mg, *p* = 0.019) and M30 (β = 30.66 mg, SE = 14.90 mg, *p* = 0.040) for out of school PA compared to the control group. In relation to time effects, M60 during the weekend decreased from baseline to post-intervention. Furthermore, at least one or more PA metrics were found to be lower at follow up compared to baseline for whole week, weekend and school time segmented periods.

### 3.4. Effects of Covariates on Physical Activity Outcomes

The intention to treat multilevel analysis results, including full models with covariates, can be found in Appendix A. The neighborhood deprivation decile index was not associated with PA in any of the segments of the week or during the whole week. Sex (boys higher PA) was significantly associated with MVPA, mean ENMO, M60 and M30. Decimal age was significantly and positively associated with increased mean ENMO during the whole week and MVPA during the weekend. Presenting special educational needs was significantly associated with decreased mean ENMO and M30 outside school. IOTF SDS BMI was significantly and negatively associated with M60 during whole week only. Significant associations were found between ethnicity and MVPA, mean ENMO, M60 and M30, respectively. Specifically, White British children presented higher levels of mean MVPA, mean ENMO and M60 during the weekend, higher mean ENMO and M60 during the whole week and, lastly, higher mean ENMO and M30 out of school. The participation in a sport event within school (e.g., school sports week) was positively associated with MVPA, mean ENMO and M30 only during school time. For environmental variables, rainfall was significantly and negatively associated with engagement in both MVPA during the whole week, within school time and outside school, it was negatively associated with mean ENMO during the whole week, weekend and outside school, while it was negatively associated with M60 during the whole week and weekend. Furthermore, percentage of daylight over a day was significantly associated with increased MVPA within all the week segments and mean ENMO within all the week segments apart from school time while it was positively associated with M60 during the weekend and M30 out of school. Mean daily temperature was positively associated with mean ENMO and M30 during school time only. Accelerometer valid wear time was significantly associated with increased MVPA and mean ENMO within all the week segments apart from school time, while wear time was positively associated with M30 out of school.

## 4. Discussion

This study aimed to evaluate the effect of linear and nonlinear pedagogy PE interventions on the PA levels of 5–6-year-old children from areas of high deprivation. Our findings suggest that participation in the linear and nonlinear pedagogy PE interventions did not lead to increased PA compared to participation in the control group. This lack of intervention effect was generally consistent across intention to treat and complete case analysis and across all PA metrics and whole week (habitual), weekend, weekday in school and weekday outside of school segmented periods for PA. These findings suggest that enhanced PE would need to be extended and supplemented by whole school approaches to PA promotion and multi-component interventions targeting home and community settings to increase PA among this population.

The results presented from the intention to treat analysis using imputed data and the complete cases analysis concerning the examination of group effects and group by time interaction effects for PA outcomes were generally similar, with some exceptions. Specifically, the complete case analysis found a significant group by time interaction effect for MVPA within school at follow-up (T2) in favor of the nonlinear pedagogy intervention, compared to control group. Significant group by time interaction effects were also observed in outside of school PA metrics at post-intervention (T1), with participation in the Linear pedagogy intervention associated with lower PA metrics, relative to the control group participants. Nevertheless, the positive intervention effect found in the nonlinear pedagogy group for MVPA in school during weekdays at follow-up was not confirmed by any other result. Furthermore, the negative intervention effect found in the Linear pedagogy group for out of school PA during weekdays at post-intervention assessments might be due to the Linear pedagogy group presenting significantly higher levels of PA compared to the control group within the complete case analysis, and therefore potential regression to the mean in this sample [93]. The differences between the intention to treat analysis and complete case analysis might also be attributed to a lack of statistical power within the complete cases analysis and the exclusion of 73 valid PA observations because of missing covariates (i.e., listwise deletion), which might have affected the results [94]. Overall, the complete case analysis did not provide strong evidence for an intervention effect on children’s PA and, therefore, the results from the intention to treat analysis were accepted as an accurate portrayal of between-group differences.

The lack of linear and nonlinear pedagogy intervention group improvements in PA outcomes is consistent with previous research that has examined the effectiveness of PE interventions on children’s habitual PA using device-based methods [26,27,28,29]. These findings are in contrast to studies employing self-report or parent proxy measures, which have generally found that PE interventions increased habitual PA levels [22,23,24,25,26,27,28,29,30,31]. Nevertheless, results from self- or parent-proxy reported PA measurement should be interpreted cautiously due to factors such as recall bias, social desirability bias and the difficulty for children in classifying PA intensities and domains [95]. In comparison to the present study, the interventions examining the effect of PE on PA using device-based methods lasted for a longer duration (i.e., between 2 and 4 years) and involved older children (i.e., children aged between 8 and 11 years) [26,27,28,29]. Furthermore, the majority of these interventions included additional intervention components outside of PE (e.g., classroom sessions), but still found no effect on PA [26,27,28,29]. To the best of our knowledge, only Manios et al. (1998) has conducted a PE intervention and examined PA amongst a similar age group (6–7 years old) [23]. Their study reported that participation in a three-year PE intervention significantly increased children’s self-reported PA. Aside from the limitations attached to using a self-report measure, the positive effects in this study may be because the intervention focused on fitness rather than movement competence and incorporated classroom-based health and nutrition sessions. It is possible that the lack of intervention effects in our study could be due to the length of the PE intervention not being sufficient to impact on PA outcomes (two lessons per week for 15 weeks). Only two studies [24,25] have reported PE interventions with a similar duration compared to our study (i.e., 5 to 12 weeks). These interventions targeted teaching behaviors and teaching styles to improve children’s motivation towards PA engagement and foster physical literacy, respectively. Both reported significant increases in self-reported PA in children but did not involve device-based PA measurements [24,25]. Recently, Lahti et al. (2018) showed that children participating in daily PE during each school day maintained increased levels of habitual PA over the years compared with children who participated in only 60 min of PE per week [96]. This suggests that a stronger dose of the SAMPLE-PE interventions may be needed to obtain positive intervention effects on children’s habitual PA levels.

This study showed that PE interventions based on different pedagogical approaches did not lead to increased PA in children compared to PE delivery that followed usual practice. Nonetheless, we consistently found variables related with increased PA, such as participation in school sport week events [84] or increased daylight [86,87]. The positive associations between PA and both participation in sport events during school and daylight duration indicates that children were more active when they had more opportunities to be active. This suggests that providing children with high quality movement experiences in PE might not be sufficient to increase children’s PA if children are not provided with more and better occasions to be active—both at school and outside school—alongside the necessary equipment [97]. For children this age, daily activities are generally dictated by adults (e.g., teachers or parents) and children have low autonomy over their activity choices. This is consistent with research showing that supporting parents in setting PA goals and planning time for their children to be physically active were generally effective in increasing children’s PA [98]. Furthermore, children from deprived areas are provided with less opportunities to be active and the neighborhood is generally not seen as a safe place for children to play without supervision [99]. Thus, it might be difficult for children to apply what is experienced in PE within different settings and contexts outside of school [100,101]. Despite the lack of intervention effects, the focus of linear and nonlinear PE interventions on movement competence may lead to higher levels of PA and sport participation in later childhood and adolescence, as the association between actual and perceived competence and PA strengthens over time [37,38,39,40,41].

When considering PA measurement at baseline, more than half of children met the PA guidelines across the whole week (65%), with around 50% of children meeting guidelines over the weekend (51%). Similarly, on weekdays more than half of the children achieved 30 min within school (60%) while slightly less than 50% achieved 30 min out of school (48%). MVPA levels over the whole week reported in this study of 5 to 7-year-old children (mean MVPA: 73.74 min, SD = 22.21) were higher than the MVPA levels observed in a large dataset of English children aged between 7 and 8 (mean MVPA = 60.6 min) [7]. This is in line with what is expected, as 5 to 6 year olds are generally more active than 7–8-year-old children [7,80]. However, as shown in Appendix A, the overall MVPA levels reported in this study for within school (mean MVPA: 36.37 min, SD = 11.59 min) and out of school (mean MVPA: 32.14 min, SD = 13.81 min) were lower than those reported in 7–11-year-old UK children during school (boys: mean MVPA = 46.1 min; girls: mean MVPA = 40.7 min) and after school (boys: mean MVPA = 49.4 min; girls: mean MVPA = 47.2 min) [82]. This could be due to our sample including children from deprived areas that might have limited PA experiences during school time as well as limited or no access to safe outdoor spaces at home or in the neighborhood, and low accessibility to community sports provision to be active out of school [102]. Nevertheless, the PA levels observed in this study during weekdays, weekend and in school (weekdays: mean MVPA = 76.91 min, SD = 22.92 min; weekend: mean MVPA = 67.84 min, SD = 28.89 min), were very similar to those reported in a review summarizing objectively measured PA in school-aged children from 4 to 18 (weekday: mean MVPA = 82.3 min, SD = 44.0 min; weekend: mean MVPA = 68.3 min, SD = 43.9 min; in school: mean MVPA = 34.4 min, SD = 14.6 min) [103]. Despite the fact that a large percentage of children in our study met the PA guidelines, we found that children’s MVPA and mean ENMO declined from baseline to follow-up during the weekend. This is consistent with previous research showing that children’s PA levels decline over time [80] and suggests that interventions should focus on preventing the age-related decline in PA, particularly at weekends.

Similar to previous literature, we found that females were consistently less active than males [81,82]; children with special educational needs were less active than other children [67]; white British children were generally more active than children from other ethnicity groups [12]; BMI was negatively associated with PA levels [83]; school sport events were associated with higher engagement in PA [84], and seasonal factors, such as daylength and mean temperature, were positively associated with PA, while rainfall was negatively associated with PA [86,87]. The lack of an association between children’s PA and neighborhood deprivation level could be due to the fact that the vast majority of the children in our sample were from deprived areas within the same deprivation decile. Nonetheless, our findings indicate that inequalities in PA levels are evident from an early age and that interventions should target subgroups for PA promotion including girls, black and ethnic minority groups, and overweight/obese children.

Based on our findings, future studies aiming at increasing PA or evaluating the effects of pedagogical approaches to PE in children within deprived areas should also find strategies to widen opportunities for children to be active. Researchers and practitioners should therefore consider a whole school and community approach where parents and schoolteachers are also involved to create better opportunities for children to be active within and outside school, together with appropriate and rich educational experiences during PE hours [104,105]. In particular, for children living in deprived areas, researchers and practitioners should consider the challenges faced by schools and families and should design solutions to overcome problems in this specific population. For example, training schoolteachers to deliver pedagogical approaches might be a more cost-effective way for schools to provide PE interventions rather than paying external coaches. Furthermore, trained schoolteachers could feasibly apply pedagogical principles outside PE, such as during playtime, during after-school activities or during school sport events, and they could more easily provide an intervention for the entire duration of a school year. Moreover, schoolteachers have a closer relationship with parents compared with external coaches and might inform them about the importance of providing children with increased PA opportunities.

This study presented several strengths, comprising the inclusion of device-based measurement of PA; the use of novel, comparable, and easy to interpret raw acceleration metrics; the inclusion of at least 3 weekdays and one weekend day as a valid week criteria to guarantee that our PA assessment is representative of children’s normal PA levels over the whole week; the inclusion of a fidelity assessment to check that interventions were delivered as expected; the presence of both imputed data and complete case analysis to better interpret the outcomes of this study; finally accounting for a wide number of covariates, including weather and seasonal variation effects on PA. Furthermore, to the best of our knowledge, this study was the first to assess the effect of different pedagogical approaches based on movement-learning theories on PA and the first study in 5–7-year-old children assessing the effects of PE interventions using device-based measures. This study also has some limitations, such as the presence of 39% missing data within PA variables due to children moving to another school, dropping out from the study, not wearing the monitor enough to obtain a valid a PA measurement, or losing the accelerometer during the assessment period. However, the amount of missing data reported in this in this study is similar to that reported in previous research using device-based measurements of PA [106,107]. A further limitation is that most of the children did not wear the monitor overnight and that there was low compliance from parents with filling in PA wear time diaries, leading to the impossibility to calculate the waking time for each individual.

## 5. Conclusions

We suggest that PE interventions, based on linear and nonlinear pedagogy, are not sufficient to increase PA levels in 5–6-year-old children, compared to common practice. Possible explanations for a lack of an intervention effect could be the short duration of the intervention, the low autonomy of children in this age group over their spare time [100,101] and the lack of actions to target barriers to PA engagement [97]. Therefore, future research should consider implementing strategies to increase occasions for children to apply the movement skills learnt during PE as well as enhanced PE sessions guided by pedagogical approaches. Furthermore, practitioners should consider more holistic approaches to supplement pedagogical approaches, such as whole school programs of PA promotion and multi-component interventions targeting home and community settings to increase PA in children where teachers and parents present an active role in creating opportunities for children to practice movement skills and be active. In particular, training schoolteachers to provide pedagogical interventions in PE could be a cost effective and viable option to increase the amount of time children are exposed to pedagogical approaches and might potentially lead to increased occasions for children to be active in schools and facilitate providing interventions for longer periods of time and could facilitate informing parents about the importance of providing children with occasions to be active.

## Figures and Tables

**Figure 1 children-08-00049-f001:**
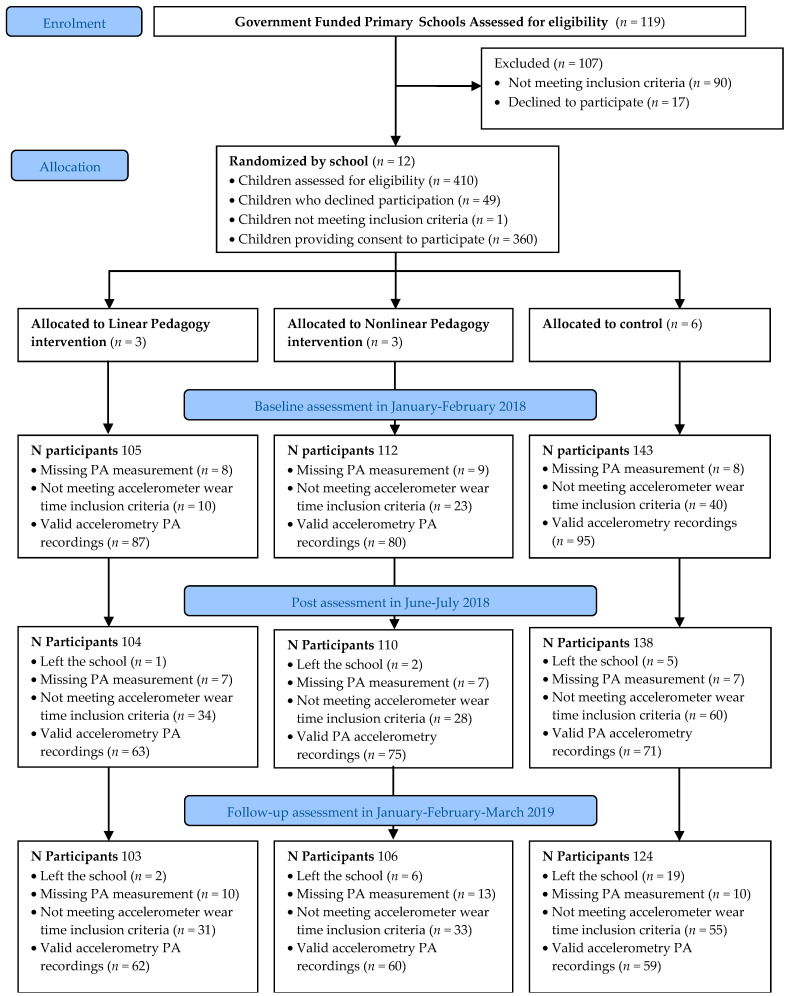
Flow diagram. PA: Physical activity.

**Table 1 children-08-00049-t001:** Baseline characteristics of children by group.

	Linear Pedagogy (*n* = 105)		Nonlinear Pedagogy (*n* = 112)		Control (*n* = 143)	
Baseline Characteristic	Mean (SD) or %	Missing Data	Mean (SD) or %	Missing Data	Mean (SD) or %	Missing Data
Decimal Age (years)	6.0 (0.3)	5	5.9 (0.3)	1	5.9 (0.3)	2
Females	53%	0	52%	0	58%	0
White British	68%	8	52%	9	50%	5
SEN	8%	1	15%	1	12%	0
Living within the 30% most deprived areas (IMD)	96%	4	77%	1	89%	3
IOTF SDS BMI	0.4 (1.3)	9	0.5 (1.1)	8	0.3 (1.1)	27
Thinness grade 3	1%		0%		1%	
Thinness grade 2	2%		1%		0%	
Thinness grade 1	6%		4%		6%	
Healthy weight	61%		67%		67%	
Overweight	21%		72%		22%	
Obese	8%		9%		4%	
*Meeting PA guidelines*						
Whole week	68%	18	64%	32	62%	48
Weekdays	70%	18	71%	32	72%	48
Weekend	53%	18	48%	32	53%	48

SD: Standard deviation; NA: missing data; SEN: Special educational needs; IMD: Index of multiple deprivation; IOTF: International Obesity Task Force; SDS: Standardized Scores; BMI: Body Mass Index; PA: physical activity.

**Table 2 children-08-00049-t002:** Pedagogical fidelity checklist results.

	Category							Global	
	Category Mean (SD)							Global Mean (SD)	
	1	2	3	4	5	6	7	1	2
Nonlinear	5.00 (0.00)	5.00 (0.00)	4.90 (0.28)	3.95 (0.78)	4.05 (0.77)	4.73 (0.41)	4.58 (0.43)	5.00 (0.00)	5.00 (0.00)
Linear	1.40 (0.64)	1.48 (0.85)	1.20 (0.41)	1.77 (0.94)	1.20 (0.41)	1.63 (0.88)	1.63 (0.75)	10.40 (0.74)	1.33 (0.82)
Control	2.10 (0.83)	2.15 (1.04)	2.19 (0.88)	1.44 (0.97)	2.33 (0.87)	2.21 (0.75)	2.50 (0.54)	2.00 (1.08)	1.92 (1.11)

**Table 3 children-08-00049-t003:** Intervention effects on whole week and weekend physical activity.

	MVPA			Mean ENMO			M60		
Predictors	β	SE	*p*	β	SE	*p*	β	SE	*p*
**WHOLE WEEK PA**									
**Intention to treat analysis**									
Group [NLP] * Time [T1]	−2.62	3.17	0.414	−1.881	2.652	0.483	−1.805	10.466	0.864
Group [NLP] * Time [T2]	1.566	3.75	0.680	0.402	2.448	0.870	3.156	11.981	0.794
Group [LP] * Time [T1]	−0.637	4.04	0.876	−0.936	2.85	0.743	−0.071	14.383	0.996
Group [LP] * Time [T2]	−2.073	3.33	0.538	−2.204	2.539	0.390	−1.692	11.085	0.879
**Complete case analysis**									
Group [NLP] * Time [T1]	−2.02	3.71	0.587	−0.32	2.90	0.913	0.28	14.03	0.984
Group [NLP] * Time [T2]	5.73	4.35	0.188	4.52	3.40	0.183	3.12	16.44	0.849
Group [LP] * Time [T1]	−1.63	4.94	0.742	−1.65	3.86	0.668	−6.98	18.67	0.708
Group [LP] * Time [T2]	−1.22	4.04	0.762	−0.80	3.16	0.799	−10.63	15.28	0.487
**WEEKEND PA**									
**Intention to treat analysis**									
Group [NLP] * Time [T1]	−2.50	4.68	0.595	−0.75	4.26	0.861	7.55	14.69	0.608
Group [NLP] * Time [T2]	1.67	5.39	0.758	2.74	4.16	0.515	7.61	14.86	0.610
Group [LP] * Time [T1]	0.64	4.88	0.897	−0.81	4.02	0.841	4.76	18.96	0.803
Group [LP] * Time [T2]	−3.91	4.87	0.426	−1.74	4.18	0.680	−13.69	14.70	0.355
**Complete case analysis**									
Group [NLP] * Time [T1]	−1.18	5.86	0.841	1.97	4.44	0.656	19.29	23.17	0.405
Group [NLP] * Time [T2]	9.41	6.57	0.152	8.71	4.97	0.080	33.42	25.96	0.198
Group [LP] * Time [T1]	0.88	7.40	0.905	−0.28	5.60	0.959	12.70	29.26	0.664
Group [LP] * Time [T2]	−0.88	5.87	0.881	−0.91	4.44	0.838	−2.41	23.19	0.917

Control group is the reference category; MVPA: moderate to vigorous physical activity; ENMO: Euclidean norm minus one; M60: minimum acceleration value in the most active hour; β: estimate; SE: standard error; *p*: *p*-value; *: interaction; T0: baseline; T1: post intervention T2: follow-up; NLP: Nonlinear Pedagogy group; LP: Linear Pedagogy group; CG: Control group; multilevel models were adjusted for decimal age, sex, International Obesity IOTF BMI z-score, special educational needs, ethnicity, school sport events, household neighborhood Index of multiple deprivation decile, valid wear time, mean rainfall, mean temperature and daylength; PA data were nested within children.

**Table 4 children-08-00049-t004:** Intervention effects on physical activity in school and out of school on weekdays.

	MVPA			Mean ENMO			M30		
Predictors	β	SE	*p*	β	SE	*p*	β	SE	*p*
**IN SCHOOL WEEKDAY PA**									
**Intention to treat analysis**									
Group [NLP] * Time [T1]	−1.56	1.55	0.318	−3.29	3.57	0.358	−14.936	13.151	0.257
Group [NLP] * Time [T2]	2.23	1.57	0.162	1.45	5.19	0.783	−3.185	15.374	0.837
Group [LP] * Time [T1]	0.81	2.27	0.724	0.71	5.02	0.887	−5.437	18.36	0.768
Group [LP] * Time [T2]	0.39	1.48	0.792	2.62	3.72	0.482	2.341	14.128	0.869
**Complete case analysis**									
Group [NLP] * Time [T1]	0.16	1.78	0.930	−0.86	4.61	0.852	−14.68	20.57	0.475
Group [NLP] * Time [T2]	**5.18**	**2.11**	**0.014**	7.42	5.46	0.174	−25.53	24.34	0.294
Group [LP] * Time [T1]	1.98	2.56	0.439	1.33	6.64	0.841	−4.73	29.59	0.873
Group [LP] * Time [T2]	2.34	2.01	0.244	5.08	5.20	0.329	−6.74	23.19	0.771
**OUTSIDE SCHOOL WEEKDAY PA**									
**Intention to treat analysis**									
Group [NLP] * Time [T1]	−2.09	2.11	0.326	−1.58	3.20	0.623	3.24	13.46	0.811
Group [NLP] * Time [T2]	−0.28	2.27	0.902	0.49	3.83	0.899	10.71	13.04	0.413
Group [LP] * Time [T1]	−4.17	2.69	0.126	−5.84	4.78	0.228	−15.90	16.16	0.327
Group [LP] * Time [T2]	−3.89	2.19	0.079	−4.30	3.73	0.253	0.11	12.67	0.993
**Complete case analysis**									
Group [NLP] * Time [T1]	−2.61	2.64	0.323	−3.37	4.19	0.421	−4.48	17.59	0.799
Group [NLP] * Time [T2]	0.61	3.07	0.844	0.52	4.87	0.916	6.64	20.43	0.745
Group [LP] * Time [T1]	−**7.74**	**3.71**	**0.037**	−**12.24**	**5.89**	**0.038**	−48.03	24.70	0.052
Group [LP] * Time [T2]	−4.41	2.94	0.134	−7.44	4.66	0.111	−23.25	19.58	0.235

Control group is the reference category; MVPA: moderate to vigorous physical activity; ENMO: Euclidean norm minus one; M30: minimum acceleration value in the most active half hour; β: estimate; SE: standard error; *p*: *p*-value; *: interaction; Vs: versus; T0: baseline; T1: post intervention T2: follow-up; NLP: Nonlinear Pedagogy group; LP: Linear Pedagogy group; CG: Control group; multilevel models were adjusted for decimal age, sex, International Obesity IOTF BMI z-score, special educational needs, ethnicity, school sport events, IMD household neighborhood Index of multiple deprivation decile, valid wear time, mean rainfall, mean temperature and daylength; data were nested within child for out of school PA and nested within child and class for within school PA.

## Data Availability

The data that support the findings of this study are available from the corresponding author [L.F.] upon reasonable request.

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
