# Peer review of "Effect of Linear and Nonlinear Pedagogy Physical Education Interventions on Children’s Physical Activity: A Cluster Randomized Controlled Trial (SAMPLE-PE)"

_children, 2021, doi:10.3390/children8010049_

Round 1

Reviewer 1 Report

Good study, thank you for the opportunity to review it. 

Here are some comments to the text for the authors to consider. 

Abstract is informative and well-composed, is fine. 

Introduction - in lines between 55-65 and later lines when the authors write about motor skills instead of foundational skill I would suggest using gross and fine motor skills destinction and expand a little more on the association of PA participation and sport skills competency (look for more details for example in 'Cognitive-motor interference during fine and gross motor tasks in children with Developmental Coordination Disorder' especially that you have in your sample children with problems.

I approaciated a well-wirtten part on explanation of linear and non-linear pedagogy with good referencing, but I wonder whether the problem of promotion and long-time engagement in leisure time PA in youth may lay elsewhere - 'Maybe it is not a goal that matter but the support they receive' ? look at this issue and come back in Discussion. 

In Methods section details and procedure are described. One thing occurs strange to me - follow-up testing took place in January-February of 2019 (before the intervention)? or should it be 2020? 

and in subsection Participants I think you should mention the sample size - number of the pupils who took part in the study (not only in the Results as this is not the results of the study). I have to admit that Flow diagram gives a reader a clear vision of what and how was done. That was a good idea.

When you used abbreviations like PA use them consequently throughout the whole text (ex. line 391, 416 ect.) and in other cases as well. 

In Discussion when authors report no effects of the intervention (which is also a finding) I would expect discussing it against the findings of other studies which you have done but - once again refer back to the lack of support that maybe critical in children and youth engagement in PA. One has to consider whether they need sport/motor competency, physical fitness or something else. Do they need goals or support? Intervention involving activity tracking devices and giving the students goals showed better effects in those, who together with the daily PA goals, received support from PE teachers. Also involvement of parents seems to have a positive impact on physical activity and health behaviours of children. I think the issue of family role in meeting the recommended PA guidelines should also be tackled a little bit in explanation of the study findings. 

Tables are clear and neat, so is the language. 

Author Response

Comment 1:

Good study, thank you for the opportunity to review it.

Here are some comments to the text for the authors to consider.

Response:

We would like to thank Reviewer 1 for their constructive and positive feedback. Our responses to specific comments are outlined below.

Comment 2:

Abstract is informative and well-composed, is fine.

Response:

Thank you.

Comment 3:

Introduction - in lines between 55-65 and later lines when the authors write about motor skills instead of foundational skill I would suggest using gross and fine motor skills destinction and expand a little more on the association of PA participation and sport skills competency (look for more details for example in 'Cognitive-motor interference during fine and gross motor tasks in children with Developmental Coordination Disorder' especially that you have in your sample children with problems.

Response:

Thank you for this suggestion. We use the term movement competence in the manuscript as the terms movement competence and motor competence are used widely and interchangeably in the literature (See “Fundamental motor skills: A systematic review of terminology” by Logan et al., 2017). Furthermore, the term movement competence is widely recognised as being more easily understood by practitioners. We recognise that gross and fine motor skills are important constructs within movement competence but have used the term foundational movement skills instead as they are defined as “goal directed movement patterns that directly and indirectly impact an individual’s capability to be physically active and that can continue to be developed to enhance physical activity participation and promote health across the lifespan”. Therefore, we consider the term foundational movement skills to be more relevant to the aims of this study and therefore we have kept this as originally stated.

We read the suggested article “'Cognitive-motor interference during fine and gross motor tasks in children with Developmental Coordination Disorder”. The article concerns the relationship between motor competence (fine and gross motor skills) and cognition in normally developing children and children with developmental coordination disorder. In the current study, we recruited children from areas of high deprivation who are potentially at risk of developmental coordination disorder but we did not assess this directly as part of eligibility criteria. Therefore, we do not consider DCD to be relevant to the introduction either. 

Comment 4:

I appreciated a well-written part on explanation of linear and non-linear pedagogy with good referencing, but I wonder whether the problem of promotion and long-time engagement in leisure time PA in youth may lay elsewhere - 'Maybe it is not a goal that matter but the support they receive' ? look at this issue and come back in Discussion.               

Response:

We thank the reviewer for the comment.

The current study aimed to address the question of whether PE affects PA specifically in young children. For this reason, the introduction focuses on what is known within these topics, contexts and this population, rather than other wider settings or strategies.

We believe that we answered the reviewer’s question ('Maybe it is not a goal that matter but the support they receive' ?) within the discussion section of the manuscript where we talked about factors that could influence children’s engagement in physical activity comprising:

  • Children being more active where they are provided with opportunities to be active (e.g. sport events in schools) 567-571;
  • Physical education interventions alone might not be sufficient to increase children’s PA if children are not provided with extended and enhanced occasions to be active both at school and outside of school 571-577;
  • Adults (parents/guardians, teachers and coaches) are gatekeepers of children’s physical activity 574-577;
  • Neighbourhood deprivation affecting physical activity in children 578-581.

Comment 5:

In Methods section details and procedure are described. One thing occurs strange to me - follow-up testing took place in January-February of 2019 (before the intervention)? or should it be 2020?

Response:

We thank the reviewer for the comment and apologise for this error. Post-intervention assessments were completed between June and July 2018, while follow-up assessments (between January and February 2019. Therefore, we have addressed the comment by modifying line 144 as follows:

“intervention period between June and July 2018, while follow-up assessments (T2) took place 6 months”.

Comment 6:

and in subsection Participants I think you should mention the sample size - number of the pupils who took part in the study (not only in the Results as this is not the results of the study). I have to admit that Flow diagram gives a reader a clear vision of what and how was done. That was a good idea.    

Response:

As we reported in lines 144-147,  “The design, conduct and reporting of this study was designed in accordance with the Consolidated Standards of Reporting Trials (CONSORT) [57]”.

Consort guidelines suggest reporting information about participants in methods section as follows:

“Participants     

4a           Eligibility criteria for participants

4b           Settings and locations where the data were collected”

Furthermore, Consort guidelines suggest reporting information about participants in Results section  as follows:

“Results

Participant flow (a diagram is strongly recommended)   

13a         For each group, the numbers of participants who were randomly assigned, received intended treatment, and were analysed for the primary outcome

13b         For each group, losses and exclusions after randomisation, together with reasons

Recruitment      

14a         Dates defining the periods of recruitment and follow-up

14b         Why the trial ended or was stopped

Baseline data    

15           A table showing baseline demographic and clinical characteristics for each group

Numbers analysed         

16           For each group, number of participants (denominator) included in each analysis and whether the analysis was by original assigned groups”

Therefore, following CONSORT, we reported participant sample size and descriptives in the Results.

Comment 7:

When you used abbreviations like PA use them consequently throughout the whole text (ex. line 391, 416 ect.) and in other cases as well.

Response:

Thank you for highlighting this error. We addressed the reviewer’s comment and used the abbreviation “PA” in lines:

90, 113, 249, 290, 415 and 637.

Comment 8:

In Discussion when authors report no effects of the intervention (which is also a finding) I would expect discussing it against the findings of other studies which you have done but - once again refer back to the lack of support that maybe critical in children and youth engagement in PA. One has to consider whether they need sport/motor competency, physical fitness or something else. Do they need goals or support? Intervention involving activity tracking devices and giving the students goals showed better effects in those, who together with the daily PA goals, received support from PE teachers. Also involvement of parents seems to have a positive impact on physical activity and health behaviours of children. I think the issue of family role in meeting the recommended PA guidelines should also be tackled a little bit in explanation of the study findings.

Response:

As we reported in lines 566-581, the daily activities in 5-6 years old children are mostly dictated by adults both in school and outside school and other barriers might negatively influence their physical activity.

Given the young age our participants (5-6 year old children), it is unclear whether they could be asked to set meaningful physical activity goals for themselves and this warrants further study. We believe that it is more important to provide young children with enjoyable, yet educational, PE experiences where they explore a range of movements and activities. We find the reviewer’s suggestion about goalsetting interesting from the perspective of parents and there is evidence for this approach from family based interventions having benefits on children’s physical activity. Therefore, we added the following sentences in lines 575-577:

“This is consistent with research showing that supporting parents in setting PA goals and planning time for their children to be physically active were generally effective in increasing children’s PA [97].”

Where we cited: “Brown, H.E.; Atkin, A.J.; Panter, J.; Wong, G.; Chinapaw, M.J.M.; van Sluijs, E.M.F. Family-based interventions to increase physical activity in children: A systematic review, meta-analysis and realist synthesis. Obes. Rev. 2016, 17, 345–360, doi:10.1111/obr.12362.”

Comment 9:

Tables are clear and neat, so is the language.      

Response:

We thank the reviewer for the comment.

Reviewer 2 Report

This is a very well written manuscript describing an interesting study.There are only a few points that should be addressed.

  1. The aim of the study is not in agreement with its title.The introduction is very well written; however, the reader gets confused when compares the title and the introduction (take into account that not every reader should be aware of the SAMPLE-PE project). I strongly believe that your title should focus on the aim of this particular manuscript and disconnect with the SAMPLE-PE project.If the authors have to keep the present title, the intorduction (incuuding the aim of the study) should be changed.
  2. The aim of the study should be clearly stated in the abstract.
  3. Check the key-words. They should not be derived from the title.
  4. What about the children that did not provide consent form? Did they participate in the PE classes of the intervention that were provided to their schools?
  5. I am not aware of the education system in England and I think that several readers are not aware too. Thus, I believe that it should be helpful the authors to explain why do they use "coaches" for the intervention? There were not physical education teachers in those schools that could be used? Even if the researchers did not want to use the physical education teachers of those schools why did they use "coaches" and not "physical education teachers"? Does coaches have a sufficient educational background?
  6. In the discussion (lines 503,504) it is reported "...It is possible that the lack of an intervention effects in our study could be due to the length of the PE intervention not being sufficient to impact on PA outcomes (2 lessons per week for 15 weeks)...". However, in the Method, the rationale for the duration of the intervention is not provided.
  7. Please, check line 510: there is a "who" not needed. 

Author Response

Comment 1:

This is a very well written manuscript describing an interesting study.There are only a few points that should be addressed.

Response:

We would like to thank Reviewer 2 for the further constructive comments provided. Our responses are outlined below.

Comment 2:

The aim of the study is not in agreement with its title. The introduction is very well written; however, the reader gets confused when compares the title and the introduction (take into account that not every reader should be aware of the SAMPLE-PE project). I strongly believe that your title should focus on the aim of this particular manuscript and disconnect with the SAMPLE-PE project.If the authors have to keep the present title, the intorduction (incuuding the aim of the study) should be changed.

Response:

We addressed reviewers comment by modifying the title as follows:

“Effect of Linear and Nonlinear pedagogy physical education interventions on children’s physical activity: a cluster randomized controlled trial (SAMPLE-PE)”

Comment 3:

The aim of the study should be clearly stated in the abstract.

Response:

Thank you for highlighting this oversight. We have addressed the reviewer’s comment in lines 16-17 by adding the following sentence:

“Therefore, this study aimed to assess how different pedagogical approaches in PE might affect children's PA.”

Comment 4:

Check the key-words. They should not be derived from the title.

Response:

We modified the keywords as follows:

“Teaching, Curriculum, Primary school, Accelerometers, Movement competence.”

Comment 5:

What about the children that did not provide consent form? Did they participate in the PE classes of the intervention that were provided to their schools?

Response:

We have clarified this by adding the following sentence in lines 134-136:

“The children who did not provide consent to participate in the research study took part in physical education lessons both in the intervention and control groups.”

Comment 6:

I am not aware of the education system in England and I think that several readers are not aware too. Thus, I believe that it should be helpful the authors to explain why do they use "coaches" for the intervention? There were not physical education teachers in those schools that could be used? Even if the researchers did not want to use the physical education teachers of those schools why did they use "coaches" and not "physical education teachers"? Does coaches have a sufficient educational background?

Response:

To address this comment, we added the following sentences in Lines 151-154:

“Given that most of the generalist primary school teachers lack the confidence and competence to effectively teach PE [58], coaches were recruited to deliver the Linear and Nonlinear Pedagogy PE interventions. This in line with current practice in primary PE in England where the majority of primary schools currently employ sports coaches from external providers to deliver PE [59].”

“58.        Morgan, P.J.; Hansen, V. Classroom teachers’ perceptions of the impact of barriers to teaching physical education on the quality of physical education programs. Res. Q. Exerc. Sport 2008, 79, 506–516, doi:10.1080/02701367.2008.10599517..”

“59.        Griggs, G. Spending the Primary Physical Education and Sport Premium: a West Midlands case study. Educ. 3-13 2016, 44, 547–555, doi:10.1080/03004279.2016.1169485.”

All coaches had a Level 2 coaching qualification from a National Governing Body of sport, which indicates that they are suitably qualified to plan and lead delivery of sessions. 

Comment 7:

In the discussion (lines 503,504) it is reported "...It is possible that the lack of an intervention effects in our study could be due to the length of the PE intervention not being sufficient to impact on PA outcomes (2 lessons per week for 15 weeks)...". However, in the Method, the rationale for the duration of the intervention is not provided.

Response:

We addressed the reviewer’s comment in line 190-191 by adding the following sentence:

“Intervention duration was chosen based on previous literature showing that interventions lasting between 6 and 15 weeks are effective in increasing children movement competence [60,61]”

“60.        Logan, S.W.; Robinson, L.E.; Wilson, A.E.; Lucas, W.A. Getting the fundamentals of movement: A meta-analysis of the effectiveness of motor skill interventions in children. Child. Care. Health Dev. 2012, 38, 305–315.

  1. Foweather, L.; Rudd, J.R. Fundamental Movement Skills Interventions. In The Routledge Handbook of Youth Physical Activity ; Brusseau, T.A., Faiclough, S.J., Lubans, D.R., Eds.; Routledge: New York, 2020; pp. 715–737.”

Comment 8:

Please, check line 510: there is a "who" not needed.

Response:

We addressed reviewer’s comment in line 563:

“increased levels of habitual PA over the years compared with children who participated in only 60”

Reviewer 3 Report

See in attach

Author Response

We would like to thank Reviewer 3 for the further constructive comments provided. Our responses are outlined below.

Comment 1:

  1. Lines 7-8 - the authors must include the postal code.

Response:

We added the postal code as suggested by the reviewer.

Comment 2:

  1. Throughout the article the authors should use square brackets when quoting an author, not round brackets, please see the instructions in the template.

Response:

Thank you for highlighting this. We addressed this comment by using Multidisciplinary Digital Publishing Institute citation style.

Comment 3

  1. The authors present in the introduction many reasons regarding the importance of physical activity, as well as its means / methods of evaluation. Indeed, questionnaires are the most tools commonly used when it comes to assessing PA, but accelerometers are more appropriate. My question for the authors is how did the students included in the research be able to assess their PA level? did everyone have this device? i.e 360 participants? I see that the authors mentioned this in lines 235-239.....it's hard for me to believe!!!!

Response:

In accelerometer-based assessment of physical activity, children are not able to assess their physical activity levels as the accelerometer devices do not display their physical activity levels as this could affect children physical activity (participant reactivity). Accelerometer PA data is downloaded by the researchers when the monitors are returned and the children are not informed about their results until at the end of the study. 

We have a large stock of accelerometers (around 150) and manage data collection by distributing accelerometers between a subsample of schools across the data collection time point. For example, we simultaneously assess PA for three schools (CONTOL, LP, NLP interventions) across ten days before the monitors are downloaded and then charged to set up for the next three schools. For brevity reasons, we have not added this information to the paper as it is not typically reported.

Comment 4:

  1. Line 57 - Why does the author number 37 appear before references 33,34,35,36, in section Introduction? I did not see these references in the text (i.e 33-36) the authors must include in this section. If Australian Curriculum Assessment and Reporting Authority, 2013; SHAPE America, 2015; UK Government, 2013; UNESCO, 2015 are the references, they should be numbered.

Response:

We addressed the comment in line 55 as follows:

“children’s development of movement competence [19,33–35]. Movement competence is hereby”

Comment 5:

  1. Also, line 94, see Rudd et, al., 2020, please put the number.

Response:

We addressed the comment in line 95 as follows:

“formation that will result in the performance of functional movement solutions [54]. Consequently”

Comment 6:

  1. The authors say: ''Information about children’s demographics (i.e. date of birth, gender, ethnicity, home postcode and special educational needs) were provided by parents or guardians within a questionnaire that was returned with the consent form'', where is this instrument or what were the questions, how many questions, what kind of items were used?

Response:

These are standard demographic questions: we do not think that adding information about this questionnaire in the paper is necessary and have not provided them for reasons of brevity. If the reviewer/editor insists that this is an important inclusion then we are happy to provide the demongraphic questionnaire template.

Comment 7:

  1. ''All measurements were taken twice'', i.e at the beginning of the experiment and after the intervention? ? it is not clearly understood, please detail.

Response:

We have clarified the data collection schedule by adding the following sentence in lines 230-233:

“Demographic outcomes were collected during baseline data collection (January-February 2018) while anthropometric and physical activity outcomes were collected during each data collection point comprising baseline, post-intervention (June-July 2018) and follow-up (January-early March 2019).”

As for anthropometric measurements, it is common practice to assess anthropometrics data at least two times and to calculate the average of the measurements during each data collection point.